# Hyperbaric Oxygen Therapy for Managing Cancer Treatment Complications: A Safety Evaluation

**DOI:** 10.3390/medicina61030385

**Published:** 2025-02-22

**Authors:** Kubra Canarslan Demir, Ahmet Ugur Avci, Munire Kubra Ozgok Kangal, Berrin Ceylan, Selcen Yusra Abayli, Ismail Ozler, Kerim Bora Yilmaz

**Affiliations:** 1Department of Undersea and Hyperbaric Medicine, Gulhane Research and Training Hospital, University of Health Sciences, 06010 Ankara, Turkey; drcanarslan@hotmail.com (K.C.D.); kubra_ozgk@hotmail.com (M.K.O.K.);; 2Department of Aerospace Medicine, Gulhane Research and Training Hospital, University of Health Sciences, 06010 Ankara, Turkey; 3Department of General Surgery, Gulhane Research and Training Hospital, University of Health Sciences, 06010 Ankara, Turkey; 4Department of Medical and Surgical Research, Institute of Health Sciences, Hacettepe University, 06010 Ankara, Turkey

**Keywords:** cancer, hyperbaric oxygen therapy, metastasis, recurrence, safety, tumor progression

## Abstract

*Background and Objectives:* Hyperbaric oxygen therapy (HBOT) has shown promise in managing complications due to cancer treatments, particularly those related to radiotherapy and surgery. Despite its clinical benefits, concerns persist regarding its potential to influence cancer progression. This study aimed to evaluate the safety and clinical outcomes of HBOT in patients with active or previously treated solid tumors. *Methods:* A retrospective analysis was conducted on patients with solid tumors who underwent at least five HBOT sessions. Comprehensive data, including patient demographics, cancer type, total number of HBOT sessions, imaging findings, and clinical outcomes (recurrence, metastasis, and mortality), were collected. Descriptive statistics and the relationship between the number of HBOT sessions and long-term cancer outcomes were analyzed. *Results:* This study included 45 patients (median age: 64 years; 60% male) who received a median of 27 HBOT sessions. At initiation, 27.9% of the patients were classified as cured, 53.5% were in remission, and 18.6% had active cancer. Over a median follow-up period of 783 days, 8.7% experienced recurrence, 2.7% had persistent active cancer, and 59.5% had no recurrence. No HBOT-related complications were observed during the course of HBOT. Statistical analyses revealed no significant correlations between the number of HBOT sessions and metastasis (*p* = 0.213) or mortality (*p* = 0.881). *Conclusions:* HBOT appears to be a safe and effective adjunctive therapy for managing complications in patients with solid tumors. No evidence was found to suggest HBOT contributes to tumor progression, recurrence, or metastasis. Future prospective studies with larger cohorts are needed to confirm these results and further evaluate the therapeutic role of HBOT in oncology.

## 1. Introduction

Cancer is the second leading cause of death globally, following cardiovascular diseases. Current estimates suggest that one in five people worldwide will develop cancer by the age of 75, and 1 in 10 will succumb to the disease. Cancer treatment typically involves surgery, chemotherapy, radiotherapy, and immunotherapy administered according to established treatment algorithms. Despite the implementation of screening programs for the early detection and prevention of various cancer types, the incidence of many cancers continues to rise—an undeniable and harsh reality. Moreover, even when evidence-based guidelines are adhered to, treatment outcomes remain suboptimal in many cases. In the search for improved cancer therapies, significant focus has been placed on innovative approaches, particularly immunotherapy protocols. Within this evolving landscape, hyperbaric oxygen therapy (HBOT) has emerged as a potential complementary treatment, and its effects are currently under investigation [1,2].

Cancer treatment protocols that demonstrate proven efficacy, offer survival benefits, and adhere to established guidelines often present significant challenges for patients. Complications arising during treatment can result in prolonged hospital stays, increased readmission rates, higher treatment costs, and a decline in quality of life. Among these challenges, the most concerning consequence of complications related to surgical and other oncological treatments is the delay in initiating adjuvant therapy [3]. Delayed adjuvant therapy can lead to the progression of cancer as a systemic disease, an increased risk of local recurrences and distant metastases, and, ultimately, a reduction in expected survival.

When discussing complications associated with cancer treatment, surgical complications are often the first to come to mind [4]. Hyperbaric oxygen therapy (HBOT) has emerged as a valuable tool in managing these complications. It is widely utilized for conditions such as flap necrosis following breast surgery, surgical wound healing disorders, and tissue damage caused by radiation [3,5,6,7,8]. HBOT has been shown to accelerate surgical wound healing, effectively control infections, and successfully treat radiation-induced tissue damage [9,10,11].

Today, HBOT is established in clinical practice as an effective approach for managing radiation-related injuries and other expected complications associated with cancer treatment [4]. This complementary therapy holds significant potential to improve outcomes and enhance quality of life for patients undergoing cancer treatment.

HBOT is a treatment modality in which patients breathe 100% oxygen within a hyperbaric chamber pressurized above atmospheric pressure (>1 ATA). This therapy increases the amount of dissolved oxygen in plasma, allowing oxygen to reach tissues independently of hemoglobin [9]. HBOT has gained prominence as an effective supportive treatment for managing complications associated with surgery, chemotherapy, and radiotherapy in cancer patients. Its benefits are primarily attributed to its ability to enhance angiogenesis and reduce edema by improving oxygen delivery to hypoxic tissues [9,12].

Radiotherapy-related complications, such as osteoradionecrosis, soft tissue necrosis, radiation cystitis, and proctitis, present serious challenges. HBOT has been shown to promote tissue healing and provide symptomatic relief in these conditions [13]. In cancer patients, HBOT is utilized not only to address complications arising from cancer treatments but also for other indications independent of cancer.

Research indicates that HBOT significantly increases tissue oxygen pressure (PO2) in both normal and cancerous tissues. Furthermore, the elevated oxygen pressure within tumors can be clinically sustained for approximately 15 to 60 min following HBOT sessions [14,15,16]. These findings underscore the potential of HBOT as a complementary therapeutic strategy for improving tissue recovery and managing complications in oncology patients.

In a survey conducted among clinicians at 179 hyperbaric medical centers, 7% of respondents indicated that they believed HBOT might have a carcinogenic effect. Additionally, 42% expressed concerns about the potential for malpractice litigation in the event of tumor reactivation or accelerated malignancy progression in patients undergoing HBOT [17].

The objective of this study was to evaluate the effects of HBOT on the clinical course of solid malignancies and to assess the safety of HBOT in patients with active or previously treated solid cancers who have undergone this therapy for various indications.

## 2. Materials and Methods

This is a descriptive observational study that was conducted at the Department of Underwater and Hyperbaric Medicine, Gulhane Research and Hyperbaric Medicine Hospital. We retrospectively analyzed the medical records of patients with active or previously treated solid cancers who underwent HBOT for any indication. Data were collected on patient demographics (age and gender), HBOT details (indication and total number of sessions), cancer characteristics (diagnosis, subtype, and clinical status as active, in remission, or cured), medications, imaging findings (before and after HBOT), clinical course during treatment, recurrence or progression of active cancer, and complications associated with the procedure. Hematological and inflammatory parameters, including hemoglobin (Hg), white blood cell count (WBC), platelet count, erythrocyte sedimentation rate (ESR), and C-reactive protein (CRP), were assessed before and after HBOT. The information was retrieved from patient files and the hospital’s electronic medical record system.

The study population included patients with active or previously diagnosed solid tumors who underwent at least five sessions of HBOT. Patients without a history of solid cancer, those with non-solid malignancies, or those who received fewer than five HBOT sessions were excluded from the analysis. All patients in the study underwent HBOT once daily at a pressure of 2.4 ATA for 2 h, with five sessions per week, receiving treatment in either a monoplace chamber (Hipertech MON-08, 2014, Istanbul, Turkey) or a multiplace chamber (Hipertech ZYRON 12, 2008, Istanbul, Turkey).

The statistical analyses were performed using the Jamovi program (version 2.4.7). Data are expressed as *n* (%), mean ± standard deviation, or median (minimum–maximum). The Shapiro–Wilk test was used to analyze the normal distribution of the continuous variables. The relationship between the number of HBOT sessions and mortality and metastasis during follow-up was evaluated using the binary logistic regression test. Hosmer–Lemeshow goodness-of-fit statistics were used to assess model fit. The changes in blood parameters before and after HBOT were analyzed using the paired Student’s t-test for normally distributed variables and the Wilcoxon test for non-normally distributed variables. A 5% Type I error level was used to infer statistical significance.

This study was performed in compliance with the ethical standards of the Declaration of Helsinki. The study received ethical approval from the Gulhane Ethics Board of the University of Health Sciences (05/24: 2024-289).

## 3. Results

A total of 45 oncological patients were treated with HBOT and included in this study. Forty-six patients met the inclusion criteria; however, one patient was excluded from the study due to missing data. The median age of the patients was 64 years (range: 32–81), with 27 (60%) males and 18 (40%) females. The types of cancer diagnosed in the patients are summarized in Figure 1. All patients received HBOT, with a median of 27 sessions (range: 5–81). The HBOT indications are presented in Table 1.

At the time of HBOT initiation, 12 patients (27.9%) were classified as cured, 23 (53.5%) were in remission, and 8 (18.6%) had active cancer. The status of cancer presence before, during, and after HBOT is summarized in Table 2. None of the patients had HBOT-related complications during the HBOT schedule.

The median follow-up duration after completing HBOT was 783 days (range: 4–3510). During the follow-up period, 11 patients (29.7%) died, with a median interval of 21 months (range: 0.5–66) between the completion of HBOT and death. Among the 45 patients, 22 (59.5%) experienced no recurrence of cancer, while recurrence was observed in three patients (8.7%), and one patient (2.7%) had persistent active cancer.

The relationship between the number of HBOT sessions and mortality during the follow-up period was not statistically significant (*p* = 0.881, RR (95% CI) = 1.00 (0.965–1.04)). Similarly, there was not a significant relationship between the number of HBOT sessions and metastasis during the follow-up period (*p* = 0.213, RR (95% CI) = 0.955 (0.889–1.03)).

The distribution of cancer types among patients who died during the follow-up period is summarized in Table 3. The longest interval between HBOT completion and death was observed in a patient with parathyroid carcinoma (39 months), while the shortest interval was noted in a patient with colon carcinoma (0.5 months).

The laboratory values before and after HBOT are presented to compare the changes in laboratory parameters following hyperbaric oxygen therapy, as shown in Table 4.

## 4. Discussion

This study evaluated the clinical course of patients with active or previously treated solid tumors who underwent HBOT for various indications and examined the safety of this treatment in this patient population. Among the patients, 59.5% experienced no recurrence, 8.7% had recurrence, and 2.7% had persistent active cancer. Importantly, no significant correlation was found between the number of HBOT sessions and mortality, recurrence, or metastasis.

The key finding of this study is the inclusion of patients with active cancer, providing valuable data on the safety and efficacy of HBOT in this high-risk group. These results address conflicting opinions in the literature regarding the potential risks of HBOT in cancer patients, particularly concerns about its role in tumor progression or metastasis. Furthermore, the absence of any HBOT-related complications observed in this study is a significant finding, underscoring the safety profile of this therapy. This highlights HBOT as not only effective but as safe, further supporting its use in cancer patients.

Clinicians may express concerns that the proliferation-enhancing effects of HBOT observed in wound tissue could similarly stimulate tumor cell growth, potentially worsening the clinical course of cancer. However, the pathophysiology of tumors differs significantly from that of wound tissue. Solid tumors typically contain regions of acute or chronic hypoxia [18]. Adaptation to this hypoxic microenvironment allows cancer cells to survive and proliferate [19].

Hypoxia has been shown to trigger angiogenesis, enhance oxygenation and cellular survival through metabolic alterations such as increased glycolysis, and upregulate genes associated with cell survival and apoptosis [20]. Additionally, hypoxia increases genetic instability, promotes invasive growth, and maintains the undifferentiated state of cells, thereby contributing to carcinogenesis [18,19]. Evidence also suggests that hypoxia plays a critical role in the development of treatment resistance in cancer cells [21]. Furthermore, hypoxic conditions are known to induce the epithelial-to-mesenchymal transition, resulting in cancers with invasive and metastatic phenotypes [22,23].

Given these insights into the relationship between tumor pathophysiology and hypoxia, concerns that HBOT-induced increases in oxygenation might exacerbate cancer progression lack sufficient evidence. Instead, the available data suggest that the unique biological dynamics of tumors may limit the applicability of such concerns.

A review conducted between 1966 and 2001 evaluated 15 clinical reports examining the effects of HBOT on cancer. These reports focused on the potential impact of HBOT on tumor recurrence and metastasis. Of these studies, 12 investigated the role of HBOT as a radiosensitizer, comparing its effects in various cancer types, such as head and neck cancers, with radiation therapy. Most of these studies reported improvements in primary tumor control with HBOT, although no significant survival advantage was observed. Additionally, 10 studies demonstrated neutral or positive outcomes regarding survival duration and metastasis incidence. While some early studies suggested that HBOT might promote tumor growth, these findings were later contradicted by larger and more controlled investigations. Furthermore, no significant evidence was found to indicate the progression of malignant tumors in patients undergoing HBOT for radiation damage or non-healing wounds. Overall, this review concluded that HBOT has a neutral or even inhibitory effect on tumor growth and metastasis, providing no support for concerns that it might promote tumor progression [24].

Bennett et al. investigated the effects of HBOT in combination with radiotherapy, reporting positive outcomes in terms of local tumor control, mortality, and local recurrence. Notably, no recurrence or metastasis was observed in their study [25]. Similarly, our study found no evidence of adverse effects related to recurrence or metastasis in cancer patients treated with HBOT.

When evaluating the effects of HBOT on angiogenesis, it is crucial to consider the biological differences between wound tissue and tumor tissue. These two tissue types exhibit distinct angiogenic responses: in wound tissue, HBOT supports cellular and vascular proliferation, promoting healing. However, in tumor tissue, the dynamics of angiogenesis are far more complex and multifaceted. Tumor angiogenic responses are influenced by factors such as tissue hypoxia, cellular signaling pathways, and the tumor microenvironment. These differences underscore the distinct biological characteristics of tumors compared to wounds [24].

As highlighted by Feldmeier et al., the angiogenic properties of tumor tissue differ significantly from those of wound tissue, which weakens the likelihood of HBOT promoting angiogenesis in cancerous tissues. Therefore, when assessing the impact of HBOT on angiogenesis, the biological structure and angiogenic responses of both tissue types must be carefully considered. These distinctions are critical for ensuring the safe and controlled use of HBOT in cancer patients, emphasizing the importance of a nuanced and evidence-based approach to its application [25].

A comprehensive review investigating the potential effects of HBOT on cancer development and tumor progression analyzed studies conducted between 1960 and 1993. This review included a Medline search, as well as an examination of hyperbaric medicine textbooks and conference proceedings. A total of 24 studies were identified, comprising 12 clinical reports, 11 animal studies, and 1 study incorporating both clinical and animal data. Among the clinical reports, 3 suggested an increase in tumor growth, whereas the remaining 10 found no evidence to support such an effect. Similarly, while 2 animal studies indicated a pro-tumorigenic effect, the other 10 did not corroborate these findings. Overall, the data did not provide strong evidence that HBOT promotes cancer growth. Furthermore, initial findings suggesting tumor growth stimulation were later refuted by more extensive and controlled studies [26].

In another study, the effects of HBOT on the proliferation of breast cancer cells were examined in vitro. The findings demonstrated that HBOT inhibited cancer cell proliferation without increasing cell death. The antiproliferative effect of HBOT was found to be time-dependent and enhanced the efficacy of chemotherapeutic agents such as melphalan, gemcitabine, and paclitaxel [27].

Animal studies have further supported the potential of HBOT as a therapeutic approach for breast cancer. These studies revealed that HBOT, as a standalone treatment, exerted a significant inhibitory effect on breast tumor growth [28,29,30,31,32]. However, reviews conducted by Feldmeier et al. and Daruwalla et al. did not find a consistent effect of HBOT on breast tumor growth [24,33]. These findings align with our study and collectively suggest that HBOT may hold promise as a supportive treatment in oncology, particularly in combination with other therapies, without promoting tumor progression or growth. In our clinical evaluation, we applied HBOT for complications related to cancer or its treatments and did not observe tumor progression. At this point, the promising results of studies utilizing HBOT in the treatment of aggressive cancers and metastatic diseases corroborate the outcomes of our study [1,34,35,36].

Moen et al. showed that HBOT induces the mesenchymal-to-epithelial transition in breast tumors, resulting in a less aggressive tumor type [28]. In the 4T1 breast tumor model, Haroon et al. found that HBOT limits the growth of large tumor cell colonies [37].

HBOT has been shown to inhibit the proliferation of breast cancer cells and enhance the efficacy of certain chemotherapeutic agents. Additionally, HBOT’s ability to promote mesenchymal-to-epithelial transition in relation to metastasis suggests that it may lead to the emergence of less aggressive tumor phenotypes in breast cancer. However, some studies have reported no significant effect of HBOT on breast tumor growth. These findings underscore the need for advanced, large-scale studies to better understand the potential benefits of HBOT in breast cancer treatment. In our study, five breast cancer patients were evaluated, of whom two succumbed during follow-up, while no recurrence or metastasis was observed in the remaining three patients. The deaths could not be attributed to HBOT. It was determined that the natural progression of the disease was the primary factor in these outcomes, given the advanced stage of the disease and high proliferation index in these patients. The literature supports the use of HBOT in breast cancer patients for managing complications such as implant loss or flap necrosis, particularly in cases where cancer is active but systemic chemotherapy has not yet been initiated. These studies have reported no adverse impact on survival outcomes [38,39,40]. This evidence, along with our findings, suggests that HBOT can be considered a safe therapeutic option for carefully selected breast cancer patients.

A study investigated the efficacy of HBOT in managing radiation-induced laryngeal chondroradionecrosis (LCRN) following laryngeal cancer treatment. Conducted retrospectively, the study analyzed 29 patients diagnosed with LCRN between 2006 and 2019, of whom 34.5% received HBOT. Among these patients, a significant improvement in Chandler grades was observed (median improvement from grade 4 to grade 2.5, *p* = 0.005). These findings suggest that HBOT can provide substantial benefits in the management of LCRN, particularly for patients with persistent and treatment-resistant symptoms [41]. Additionally, a review focusing on the treatment of late radiation damage found no evidence that HBOT administered after radiation therapy for laryngeal cancer increases the risk of metastasis or recurrence [11].

Another study evaluated the growth of SQ20B and Detroit 562 head and neck squamous cell carcinoma tumors under HBOT conditions. HBOT was applied at 2.4 absolute atmospheres (ATA) for 90 min, five times per week, following a single dose of radiation. Immunohistochemical analyses were conducted to examine the effects of HBOT on tumor hypoxia and vascular structure. The results demonstrated that HBOT improved tumor oxygenation during treatment but did not promote tumor growth in either irradiated or non-irradiated tumors. Furthermore, HBOT did not increase tumor vascular endothelial growth factor (VEGF) expression, enhance vascularization, or induce lasting changes in the tumor microenvironment [42].

The findings from these studies support the notion that HBOT is a safe and effective method for managing the late-stage complications of radiotherapy. In the treatment of LCRN following laryngeal cancer, HBOT demonstrated significant improvement, particularly in patients with persistent and treatment-resistant symptoms. Moreover, HBOT administered after radiation therapy was not associated with an increased risk of cancer recurrence or metastasis. Evaluations in head and neck tumor models revealed that HBOT did not promote tumor growth or induce lasting changes in the tumor microenvironment. These results highlight the potential of HBOT to be safely applied in post-cancer treatment recovery without contributing to adverse oncological outcomes. In our study, all five patients with laryngeal cancer succumbed to their disease. HBOT was administered to these patients due to treatment-related complications, and all were in advanced stages of cancer with a poor prognosis expected. Importantly, no disease progression attributable to HBOT was observed in this cohort, further supporting its safety in this context.

Daruwalla and colleagues published two studies investigating the effects of HBOT on in vivo colon tumor models. In the first study, HBOT was assessed as a standalone treatment, and the results showed no evidence of tumor growth stimulation or the promotion of distant metastasis. Based on these findings, the researchers concluded that HBOT could be safely administered alongside other cancer treatments [43]. In the second study, HBOT was evaluated both as a monotherapy and in combination with styrene maleic acid (SMA)–pirarubicin in a primary colon cancer model. The findings demonstrated that HBOT alone did not exert any significant effect on tumors. However, when combined with SMA–pirarubicin, HBOT reduced liver metastases, inhibited tumor growth, and induced higher levels of necrosis [44]. These studies suggest that while HBOT may not directly affect colorectal cancer as a monotherapy, it holds promise as an adjunctive treatment option.

Similarly, in our study, two colorectal cancer patients underwent HBOT, and no recurrence or metastasis was observed during follow-up. This aligns with the existing evidence, further supporting the safety of HBOT in this context.

Another study explored the potential effects of HBOT on prostate cancer in a murine model of radiation-induced hemorrhagic cystitis. In this experiment, human prostate cancer cell lines (LNCaP) were injected into immunocompromised mice, and 24 tumor-free mice were randomly assigned to receive HBOT or normobaric air treatment. The results showed that HBOT did not accelerate prostate tumor growth, with no significant differences observed in tumor development between the HBOT and control groups. Furthermore, the tumor invasion, necrosis, microvascular density, and proliferative indices were similar in both groups. These findings support the notion that HBOT does not increase the risk of prostate cancer recurrence and can be safely used for the treatment of radiation-induced hemorrhagic cystitis [45].

Consistent findings from other studies have also reported no differences in in vivo tumor growth after HBOT, with no changes observed in parameters related to tumor pathology, such as microvascular density, cell differentiation, cell proliferation, or apoptosis [46,47]. Kalns et al. investigated the effects of HBOT on two different prostate cancer cell lines in vitro. Both cell lines were exposed to 100% oxygen at 3.0 ATA for 90 min, and compared to normobaric controls, their growth rates decreased by 8.1% and 2.7%, respectively [48]. In our study, six prostate cancer patients underwent HBOT, and no recurrence or metastasis was observed during follow-up. These findings further confirm that HBOT does not have a negative impact on prostate cancer or trigger recurrence. Therefore, HBOT can be considered a safe therapeutic option, particularly for conditions such as radiation-induced hemorrhagic cystitis.

In a comprehensive review, Daruwalla et al. noted that HBOT, when combined with radiotherapy, might increase tumor aggressiveness in some cases. However, larger studies involving broader cohorts of bladder cancer patients found no significant impact of HBOT on tumor progression when used alongside radiotherapy [33]. Our study aligns with these findings, as only one patient with a history of bladder cancer was included, and no recurrence or metastasis was observed during follow-up. This study is limited by the heterogeneity of the included cancer types, which have varying pathological characteristics, behaviors, and biological mechanisms, as well as the small sample size, which restricts the generalizability of our findings. Given these limitations, these findings should be regarded as preliminary results that require further validation in larger, more homogeneous cohorts. Additionally, we chose not to perform Kaplan–Meier analysis due to the substantial heterogeneity and small sample size, although Kaplan–Meier analysis is generally recommended for mortality assessment. Overall, the evidence indicates that HBOT is a safe and effective treatment option for managing complications related to cancer therapies without promoting tumor progression or metastasis.

## 5. Conclusions

HBOT was shown to be a safe and effective adjunctive therapy for managing complications in patients with solid tumors. No evidence was found to suggest HBOT contributes to tumor progression, recurrence, or metastasis. Future studies with larger, stratified cohorts and control groups are needed to evaluate the differential effects of HBOT on specific cancer types.

## Figures and Tables

**Figure 1 medicina-61-00385-f001:**
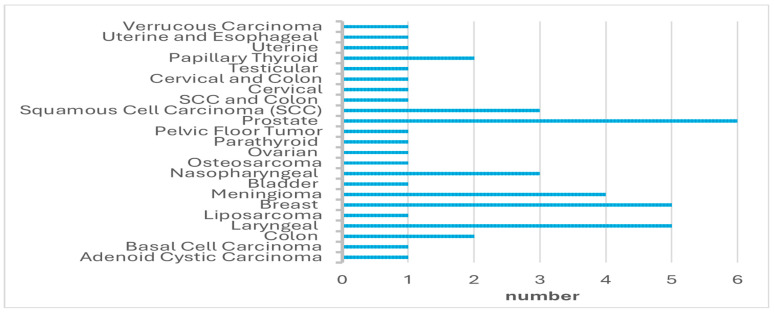
The types of cancer.

**Table 1 medicina-61-00385-t001:** The HBOT indications of patients.

Indications	Indication Subgroup	*n* (%)
Wound healing	Post-operative complicated wound healing (*n* = 10)	16 (35.6%)
	Chronic wound (*n* = 5)	
	Diabetic foot (*n* = 1)	
Delayed radiation injury	Radiation cystitis (*n* = 6)	15 (33.3%)
	Late complication after radiotherapy (*n* = 4)	
	Unspecified effects of radiation (*n* = 3)	
	Chronic radiodermatitis (*n* = 2)	
Avascular necrosis		4 (8.9%)
Compromised grafts and flaps		3 (6.7%)
Chronic osteomyelitis		3 (6.7%)
Intracranial abscess		2 (4.4%)
Central retinal artery occlusion		1 (2.2%)
Sudden sensorineural hearing loss		1 (2.2%)
Total		45 (100%)

**Table 2 medicina-61-00385-t002:** Cancer status before and after HBOT sessions.

Category	Condition	*n* (%)
Metastasis (pre-HBOT)	Yes	4 (9.5%)
	No	38 (90.5%)
Recurrence (pre-HBOT)	Yes	6 (14.3%)
	No	36 (85.7%)
Metastasis (post-HBOT)	Yes	2 (7.4%)
	No	25 (92.6%)
Recurrence (post-HBOT)	Ongoing active cancer	1 (3.7%)
	Yes	3 (11.1%)
	No	23 (85.2%)
Mortality	Yes	12 (26.6%)
	No	33 (73.3%)

**Table 3 medicina-61-00385-t003:** Mortality and recurrence during follow-up period after HBOT by cancer type.

	Deceased	Alive	Ongoing Active Cancer	With Recurrence	Without Recurrence	Total * *n* (%)
Adenoid cystic carcinoma	0	1	0	0	1	1 (2.2%)
Basal cell carcinoma	0	1	0	0	1	1 (2.2%)
Colon	1	1	0	0	2	2 (4.4%)
Laryngeal	5	0	0	0	0	5 (11.1%)
Liposarcoma	1	0	0	1	0	1 (2.2%)
Breast	2	3	0	0	3	5 (11.1%)
Meningioma	0	4	0	0	2	4 (8.9%)
Bladder	0	1	0	0	0	1 (2.2%)
Nasopharyngeal	0	3	1	1	1	3 (6.7%)
Osteosarcoma	0	1	0	0	1	1 (2.2%)
Ovarian	0	1	0	0	1	1 (2.2%)
Parathyroid	1	0	0	0	0	1 (2.2%)
Pelvic floor tumor	1	0	0	0	0	1 (2.2%)
Prostate	1	5	0	0	4	6 (13.3%)
Squamous cell carcinoma (SCC)	0	3	0	1	2	3 (6.7%)
SCC and colon	0	1	0	0	1	1 (2.2%)
Cervical and colon	0	1	0	0	1	1 (2.2%)
Cervical	0	1	0	0	1	1 (2.2%)
Testicular	0	1	0	0	0	1 (2.2%)
Papillary thyroid	0	2	0	0	1	2 (4.4%)
Uterine	0	1	0	0	0	1 (2.2%)
Uterine and esophageal	0	1	0	0	0	1 (2.2%)
Verrucous carcinoma	0	1	0	0	1	1 (2.2%)
Total	12	33	1	3	23	45 (100% **)

* Total number of deceased and alive patients. ** Due to rounding, some totals may not correspond with sum of separate figures.

**Table 4 medicina-61-00385-t004:** Laboratory values before and after HBOT.

	Pre-HBOT	Post-HBOT	*p*-Value
Hemoglobin (g/dL)	11.9 (7.7–15.2)	12.1 (4.2–16)	0.629
White blood cell (10^3^ cell/µL)	7.3 (4–18)	7.65 (4–12)	0.541
Platelet (10^3^ cell/µL)	265 ± 91.7	261 ± 91.2	0.714
Sedimentation (mm/h)	52.2 ± 34.3	48.8 ± 33.1	0.586
C-reactive protein (mg/L)	24 (1.4–300)	9.80 (0.4–126)	0.006

## Data Availability

Data are available upon reasonable request.

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
