# Peer review of "Hyperbaric Oxygen Therapy for Managing Cancer Treatment Complications: A Safety Evaluation"

_medicina, 2025, doi:10.3390/medicina61030385_

Round 1
Reviewer 1 Report
Comments and Suggestions for Authors
This is a small but interesting study, evaluating the safety and efficacy profile of hyperbaric oxygen therapy in patients with a history of cancer. In particular, the dual evaluation of efficacy in improving symptoms associated with radiotherapy and other complications (efficacy) with the risk of HBOT-induced complications is of interest.
The title is biased, as it leads the reader to the conclusion desired by the researchers. It is suggested to change it to "Hyperbaric Oxygen Therapy: Safe Evaluation in Managing Cancer Treatment Complications"
The Abstract is short, clear and simple. It presents the main information of each section. It shows objective results with numerical variables. It does not present important biases. The conclusion includes a sentence with an excessive statement (These findings highlight its potential as a supportive treatment.) that could be deleted.
The introduction is quite long, it would be advisable to reduce it, since it provides too much information. Furthermore, some statements are somewhat vague and without bibliographical support ("Clinicians have expressed concerns that the increase in PO2 in tumor tissues during HBOT could potentially promote cell proliferation not only in tumor tissue but also in scar tissue. Consequently, when considering HBOT for patients with malignancies, the possibility that the treatment could contribute to metastasis or recurrence is often taken into account. This concern has, in some cases, led to patients being denied access to HBOT.") so it is suggested that a reference be provided or that the sentence be deleted.
The materials and methods are described clearly and concisely, with sufficient data to ensure reproducibility.
It would be good to add the characteristics of the hyperbaric pressure chamber used (make, model, etc.).
It would also be good to describe laboratory controls that may have been performed on the patients.
The results are expressed clearly, quite simply, and a simple comparative analysis is performed using basic statistical tests. The characteristics of the sample and the data obtained do not seem to allow for a more in-depth analysis. It would be good to include parameters of some collateral variable that shows the effect of hyperbaria: were laboratory tests performed before and after the sessions? Was arterial saturation of the patients performed?
The discussion is very extensive, considering results from other studies, population comparisons, factors that could have affected the results, etc. It is clear that the results obtained and the possible confounders were analyzed in detail. A good analysis of the limitations of the study is carried out.
The conclusion should be short, since it is excessively extensive, includes statements that are not a consequence of the study but mere arguments and personal opinions, and they make us lose focus on the true findings that should be reduced to: HBOT is shown to be safe and effective adjunctive therapy for managing complications in patients with solid tumors. No evidence was found to suggest HBOT contributes to tumor progression, recurrence, or metastasis. Future prospective studies are needed.
Author Response
Responses 1:
We thank the reviewers for their valuable comments.
Comments 1: The title is biased, as it leads the reader to the conclusion desired by the researchers. It is suggested to change it to "Hyperbaric Oxygen Therapy: Safe Evaluation in Managing Cancer Treatment Complications"
Response 1: We agree with this comment. The title has been simplified in accordance with the reviewers' suggestions.
Comments 2: The Abstract is short, clear and simple. It presents the main information of each section. It shows objective results with numerical variables. It does not present important biases. The conclusion includes a sentence with an excessive statement (These findings highlight its potential as a supportive treatment.) that could be deleted.
Response 2: In the Abstract section, the sentence highlighted as insufficiently supported and overly assertive by the reviewers has been removed.
Comments 3: The introduction is quite long, it would be advisable to reduce it, since it provides too much information. Furthermore, some statements are somewhat vague and without bibliographical support ("Clinicians have expressed concerns that the increase in PO2 in tumor tissues during HBOT could potentially promote cell proliferation not only in tumor tissue but also in scar tissue. Consequently, when considering HBOT for patients with malignancies, the possibility that the treatment could contribute to metastasis or recurrence is often taken into account. This concern has, in some cases, led to patients being denied access to HBOT.") so it is suggested that a reference be provided or that the sentence be deleted.
Response 3: In line with the reviewers' suggestions for the Introduction section, unsupported parts have been removed, and the section has been shortened.
Commentes 4: The materials and methods are described clearly and concisely, with sufficient data to ensure reproducibility.
It would be good to add the characteristics of the hyperbaric pressure chamber used (make, model, etc.).
Response4: Thank you for your valuable feedback. The characteristics of the hyperbaric pressure chamber used (make, model, etc.) have been added.
Thank you for your valuable feedback. In our routine practice, we do not measure arterial saturation during HBOT sessions because the patients breathe 100% oxygen under increased atmospheric pressure, ensuring adequate oxygenation. The hyperbaric oxygen therapy protocol we follow is well-established and widely used, and monitoring arterial saturation is not considered necessary unless specific clinical concerns arise, as the therapy inherently provides sufficient tissue oxygenation. Additionally, there are no international guidelines recommending routine monitoring of arterial saturation during HBOT. However, in our clinic, we perform arterial saturation monitoring for intubated or critically ill patients. Since all the patients included in our study had vital signs within normal ranges, such monitoring was not required
Comments 5: It would also be good to describe laboratory controls that may have been performed on the patients.
The results are expressed clearly, quite simply, and a simple comparative analysis is performed using basic statistical tests. The characteristics of the sample and the data obtained do not seem to allow for a more in-depth analysis. It would be good to include parameters of some collateral variable that shows the effect of hyperbaria: were laboratory tests performed before and after the sessions? Was arterial saturation of the patients performed?
Respose 5: I have included the parameters of laboratory tests performed before and after the sessions in Table 4 of the Results section.
Comments 6: The discussion is very extensive, considering results from other studies, population comparisons, factors that could have affected the results, etc. It is clear that the results obtained and the possible confounders were analyzed in detail. A good analysis of the limitations of the study is carried out.
Response 6: We thank the reviewers for their valuable contributions and comments.
Comments 7: The conclusion should be short, since it is excessively extensive, includes statements that are not a consequence of the study but mere arguments and personal opinions, and they make us lose focus on the true findings that should be reduced to: HBOT is shown to be safe and effective adjunctive therapy for managing complications in patients with solid tumors. No evidence was found to suggest HBOT contributes to tumor progression, recurrence, or metastasis. Future prospective studies are needed.
Response 7: The Conclusion section has been shortened and simplified based on the reviewers' critiques. It has been revised to better align with the main message of the study.

Reviewer 2 Report
Comments and Suggestions for Authors
The manuscript is well structured and well written (except for some linguistic nuances that could be improved). However, there are some problems that should be taken into account by the authors.
MINOR PROBLEMS
They say that HBOT has established itself in clinical practice as an effective approach for managing radiation-related injuries and other expected 84 complications associated with cancer treatment. However, they rely on a single literature reference to make this claim (Fernandez E, Morillo V, Salvador M, et al; 2021). We believe that the level of scientific evidence is scarce (from a quantitative and qualitative point of view), so it would be desirable that more articles with a greater impact were substantiated (if there were a systematic review or meta-analysis it would be better or at least show more original and more recent articles).
In the Materials and Methods section, it is not clear what the study design was. It appears that they have evaluated a series of cases. That is, there is only one study group. However, in the last paragraph they say that for quantitative variables they performed Student's t-test (or Mann Whitney U test) and for qualitative variables they performed Chi-Square (or Fisher Exact test), but these tests would be used in the case of comparing two groups of elements (and the authors have evaluated a single group). There is also a problem regarding the statistical test applied when talking about mortality. To assess mortality (or survival) it would be appropriate to use Kaplan-Meier survival curves. If we look for example at what they say in the results section (line 175 and following), if they are looking for relationships between a dichotomous qualitative variable as a dependent variable (metastasis YES versus NO) versus a quantitative variable (number of sessions), the correct thing to do would be to apply a binary logistic regression. If we have three or more variables (as in the case of recurrence) a multinomial logistic regression should be carried out.
All this should be corrected, or they would have to justify what they have actually done and justify it so that it is correct.
Table 1 does not have a good form of presentation. Since the categories are the same, the values in columns 3 and 4 should be below columns 1 and 2 (instead of adding two more columns). We assume that the authors have presented it this way to reduce the size and optimise space. In any case, the authors could even consider rewording the information in the tables (put it as text), as it might take up less space. It would be desirable that each time they present data of this type, N (sample size or total number of subjects) should appear.
In table 3, it would be useful to have an additional last row showing the total (the sum) of the values of each of the four categories (Deceased, Alive, Ongoing active cancer, With recurrence, Without recurrence) and to put the percentage in brackets next to the total sum. In this table 3 there are errors. For example, if we add the values for mortality (which is in the last row and indicates yes versus no), the total number of patients (i.e. 45 people) should be the result, but 44 appear.
MAJOR PROBLEMS
The most serious methodological problem is that they are taking an initial approach that is not entirely correct. That is, they are grouping together very different types of cancer, from very different tissues, with very different pathological characteristics and very different behaviour (in terms of tumour biology). This heterogeneity does not allow the different types of cancer to be grouped together.
The sample size (45 subjects) does not allow stratification into 14 tumour types as the maximum in some categories is 5 subjects (which may correspond to different histological types and different tumour grades or stages) and some categories have only one subject (such as parathyroid or pelvic floor cancer). These sample sizes do not allow statistical inferences to be made or conclusions to be drawn that can be extrapolated to the general population.
The initial approach is not correct, and the sample size is too small to extrapolate to the general population to make inferences or conclusions, and the statistical studies are incorrect.
As possible alternatives or options, we propose some ideas to the authors:
1) Expand the sample size but focus on a single tumour type. After that, recruit a control group that is not administered HBOT but has the same characteristics as the sample of cancer patients reported in this study. By having two groups they could study very specific variables and compare them with each other in terms of survival in one group and the other. They could also compare some of the variables they have assessed in this manuscript.
2) Enlarge the sample size enough so that the statistical power would allow intra-group comparisons to be made regarding the differences between one type of cancer and the other (for example, whether cancer A responds better than cancer B in terms of survival after adjuvant HBOT treatment). However, this would not be of much value if these differences were also present in the absence of HBOT treatment.
In both option 1) and option 2), a sample size calculation should be done beforehand, but taking into account that tumour types with such different characteristics should not be aggregated.
Comments on the Quality of English LanguageIt needs a revision from a native speaker.
Author Response
Responses 2:
We thank the reviewers for their valuable comments.
The article has been revised in terms of language and fluency as suggested by the referee.
MINOR PROBLEMS
Comments 1: They say that HBOT has established itself in clinical practice as an effective approach for managing radiation-related injuries and other expected 84 complications associated with cancer treatment. However, they rely on a single literature reference to make this claim (Fernandez E, Morillo V, Salvador M, et al; 2021). We believe that the level of scientific evidence is scarce (from a quantitative and qualitative point of view), so it would be desirable that more articles with a greater impact were substantiated (if there were a systematic review or meta-analysis it would be better or at least show more original and more recent articles).
Response 1: Thank you for your insightful comment. We acknowledge the importance of providing a more comprehensive literature background on this topic. In response, we have incorporated additional references in the introduction and discussion sections to present a more balanced perspective on the current evidence regarding HBOT for radiation-induced injuries.
However, we have not directly utilized these studies in our discussion, as our manuscript focuses on a different aspect of HBOT application. Instead, we have cited them where relevant to ensure that our statement about the clinical use of HBOT is better substantiated. The following systematic reviews and meta-analyses have been added to strengthen the background of our study:
- Meier et al. (2023) – A systematic review on HBOT for late radiation toxicity in breast cancer patients.
- Quah et al. (2024) – A systematic review and meta-analysis on adjunctive modalities for preventing osteoradionecrosis.
- Geldof et al. (2022) – A systematic review on HBOT for late radiation-induced tissue toxicity in gynecological cancer patients.
- Lin et al. (2023) – A Cochrane systematic review on HBOT for late radiation tissue injury.
These references provide a more extensive overview of existing research while maintaining the focus of our manuscript. We appreciate your suggestion, as it has allowed us to refine our literature support and ensure a more balanced discussion.
Comments 2: In the Materials and Methods section, it is not clear what the study design was. It appears that they have evaluated a series of cases. That is, there is only one study group. However, in the last paragraph they say that for quantitative variables they performed Student's t-test (or Mann Whitney U test) and for qualitative variables they performed Chi-Square (or Fisher Exact test), but these tests would be used in the case of comparing two groups of elements (and the authors have evaluated a single group). There is also a problem regarding the statistical test applied when talking about mortality. To assess mortality (or survival) it would be appropriate to use Kaplan-Meier survival curves. If we look for example at what they say in the results section (line 175 and following), if they are looking for relationships between a dichotomous qualitative variable as a dependent variable (metastasis YES versus NO) versus a quantitative variable (number of sessions), the correct thing to do would be to apply a binary logistic regression. If we have three or more variables (as in the case of recurrence) a multinomial logistic regression should be carried out.
All this should be corrected, or they would have to justify what they have actually done and justify it so that it is correct.
Response 2:
Thank you for your valuable comments. We understood your significant concern.
- We added the information about the study design in the manuscript. Our study is a retrospective descriptive observational study. We analyzed the follow-up (“observe”) records of oncologic patients who underwent HBOT without changing their environment.
This is a descriptive observational study which was conducted at the Department of Underwater and Hyperbaric Medicine, Gulhane Research and Hyperbaric Medicine Hospital. We analyzed the medical records of patients with active or previously treated solid cancers who underwent HBOT for any indication, retrospectively.
- As you said, we have just one group of patient; “oncological patients who underwent HBOT”. We presented their descpritional statistics (eg. demographic data, cancer types, cancer, total HBOT session number, follow-up records on cancer status; exitus, metastasis, remission).
As you noticed, we only applied further statistical tests for;
1-the relationship between HBOT session number (continuous variable) and death during follow-up (yes / no)
2- the relationship between HBOT session number (continuous variable) and metastasis during follow-up (yes / no)
3- the relationship between HBOT session number (continuous variable) and cancer recurrence during follow-up (new recurrence / no recurrence with remission / active cancer)
Our research question was “Is there a significant difference in HBOT sessions numbers in patients who died and not died?” We had defined our dependent variable as death (dicohotomous variable) and independent variable as hbot session number (continuous variable).HBOT session number data was not normally distributed, so we had chosen Mann-Whitney U test for the first two analyzes; non-parametric test. For the 3rd analyzes we had chosen Kruskal Wallis test; non parametric test.
As you noticed, we had classified death as independent variable and HBOT session number as dependent variable, mistakely. (You may find how we chosed Mann Whitney U mistakely in the Figure 1 below. We demonstrated with red arrows.) Thank you for your attention. According to your suggestion, we correctted the dependent and independent variables; classified death as dependent variable (outcome variable) and HBOT session number as independent variable (input variable). Now, we should choose binary logistic regression test. We showed with blue arrows in the Figure 1 below.
Reference: https://www-healthknowledge-org-uk.translate.goog/public-health-textbook/research-methods/1b-statistical-methods/parametric-nonparametric-tests?_x_tr_sl=en&_x_tr_tl=tr&_x_tr_hl=tr&_x_tr_pto=tc
In this respect, we revised our statistical analyses and applied logistic regression test. The relationship between HBOT session number with mortality and metastasis during follow-up was evaluated with Binary Logistic Regression test Hosmer-Lemeshow goodness of fit statistics were used to assess model fit. The relationship between HBOT session number and recurrence was evaluated by Multinominal Logistic Regression test. However, model was not fit according to goodness of fit test for multinominal logistic regression. We did not add this test results in the manuscript. You may find the revised material-method section below;
The relationship between HBOT session number with mortality and metastasis during follow-up was evaluated with Binary Logistic Regression test. Hosmer-Lemeshow goodness of fit statistics were used to assess model fit.
The relationship between HBOT session numbers and mortality during follow up period was not statistically significant.(p=0.881, RR(95% CI) 1.00 (0.965 – 1.04)). Similarly there was not a significant relationship between HBOT session number and metastasis during follow up (p=0.213, RR(95% CI) 0.955 (0.889 – 1.03)).
Finally, you recommended to use Kaplan Meier curves for mortality. As you noticed, our sample consists several different cancers (we classified them as solid tumors). Thus, many different factors (histological type of tumours, grades of tumors, co-morbidities, other therapies) have influence on patients’ outcome. There are many significant confounders. We had hesitated to apply further statistical tests on this group, however “hyperbaric medicine physicians” have a significant concern for oncological patients about HBOT safety on tumors. There is no scientifc data about HBOT safety in oncological patients. The Kaplan-meier or cox regression analysis models might not be scientifically represent mortality along with numerous confounders. We chose not to perform Kaplan Meier test in this study group.
Comments 3: Table 1 does not have a good form of presentation. Since the categories are the same, the values in columns 3 and 4 should be below columns 1 and 2 (instead of adding two more columns). We assume that the authors have presented it this way to reduce the size and optimise space. In any case, the authors could even consider rewording the information in the tables (put it as text), as it might take up less space. It would be desirable that each time they present data of this type, N (sample size or total number of subjects) should appear.
In table 3, it would be useful to have an additional last row showing the total (the sum) of the values of each of the four categories (Deceased, Alive, Ongoing active cancer, With recurrence, Without recurrence) and to put the percentage in brackets next to the total sum. In this table 3 there are errors. For example, if we add the values for mortality (which is in the last row and indicates yes versus no), the total number of patients (i.e. 45 people) should be the result, but 44 appear.
Response 3:
We sincerely appreciate your detailed feedback regarding the presentation of Tables 1 and 3. Below, we outline the revisions we have made in response to your valuable suggestions.
Table 1:
We acknowledge your concern regarding the format of Table 1. In response:
- We have restructured the table so that the values in columns 3 and 4 are now positioned below columns 1 and 2, as suggested.
- Additionally, we have ensured that the sample size (N) is explicitly stated to enhance clarity and transparency.
Table 3:
We have made the following changes to Table 3 in accordance with your recommendations:
- A final row has been added to display the total values for each category (Deceased, Alive, Ongoing active cancer, With recurrence, Without recurrence), with the corresponding percentages in brackets.
- We have carefully reviewed and corrected an inconsistency in the mortality data, where the total patient count should have been 45 but previously appeared as 44. This issue has now been resolved
MAJOR PROBLEMS
Comments 4: The most serious methodological problem is that they are taking an initial approach that is not entirely correct. That is, they are grouping together very different types of cancer, from very different tissues, with very different pathological characteristics and very different behaviour (in terms of tumour biology). This heterogeneity does not allow the different types of cancer to be grouped together.
The sample size (45 subjects) does not allow stratification into 14 tumour types as the maximum in some categories is 5 subjects (which may correspond to different histological types and different tumour grades or stages) and some categories have only one subject (such as parathyroid or pelvic floor cancer). These sample sizes do not allow statistical inferences to be made or conclusions to be drawn that can be extrapolated to the general population.
Response 4:
Thank you for your valuable feedback regarding the heterogeneity of cancer types included in our study. While we acknowledge the challenges associated with grouping cancers with varying pathological characteristics, behaviors, and biological mechanisms, we believe this approach remains valid within the scope and objectives of our research. The primary objective of our study was to evaluate the safety profile of HBOT in a heterogeneous population of patients with solid tumors, regardless of tumor type. This inclusive approach allowed us to establish a general safety baseline and reflects the real-world clinical application of HBOT across diverse oncological indications.
We agree that the heterogeneity of cancer types and the small sample size are limitations that restrict the generalizability of our findings. However, statistical analyses were conducted to account for variability within the dataset, and the consistent trends observed reinforce the safety of HBOT across all included cancer types. Additionally, the study highlights HBOT’s broad applicability as a supportive therapy for managing complications associated with cancer treatments.
In our revised manuscript, we have addressed this limitation by including the statement:
"This study is limited by the heterogeneity of included cancer types, which have varying pathological characteristics, behaviors, and biological mechanisms, as well as the small sample size, which restricts the generalizability of our findings."
We have added the following statement to the Conclusion section:
"Future studies with larger, stratified cohorts and control groups are needed to evaluate the differential effects of HBOT on specific cancer types."
We have also emphasized the importance of future studies focusing on larger, stratified cohorts to evaluate the differential effects of HBOT on specific cancer subtypes.
Comments 5: As possible alternatives or options, we propose some ideas to the authors:
1) Expand the sample size but focus on a single tumour type. After that, recruit a control group that is not administered HBOT but has the same characteristics as the sample of cancer patients reported in this study. By having two groups they could study very specific variables and compare them with each other in terms of survival in one group and the other. They could also compare some of the variables they have assessed in this manuscript.
2) Enlarge the sample size enough so that the statistical power would allow intra-group comparisons to be made regarding the differences between one type of cancer and the other (for example, whether cancer A responds better than cancer B in terms of survival after adjuvant HBOT treatment). However, this would not be of much value if these differences were also present in the absence of HBOT treatment.
In both option 1) and option 2), a sample size calculation should be done beforehand, but taking into account that tumour types with such different characteristics should not be aggregated.
Response 5:
Thank you for your constructive feedback and the alternative suggestions regarding sample size and study design. While we acknowledge the importance of larger sample sizes and the inclusion of control groups for more robust statistical analyses, we would like to explain the challenges that limited the scope of our study.
In the past, there was a widespread belief that HBOT could potentially worsen cancer prognosis by promoting tumor growth or metastasis. This misconception led to significant skepticism among both patients and clinicians regarding the safety of HBOT in oncology settings. Consequently, the inclusion of cancer patients in HBOT programs has historically been limited due to these concerns. Even though recent studies, including ours, have demonstrated that HBOT does not contribute to tumor progression, recurrence, or metastasis, this historical bias has had a lasting impact on patient recruitment and clinical practice.
Despite being one of the largest and most established hyperbaric oxygen therapy centers in Turkey, the number of cancer patients treated with HBOT in our institution remains limited. The patient cohort included in this study represents the maximum number of eligible cases we could analyze retrospectively based on our clinical records. Increasing the sample size further would require a prospective, multicenter approach, which was beyond the scope of this study.
We have emphasized this limitation in our revised manuscript and acknowledged the need for future studies with larger, stratified cohorts and control groups to evaluate the differential effects of HBOT on specific cancer types. Nonetheless, we believe our study provides valuable preliminary data on the safety profile of HBOT in oncology and lays a foundation for addressing the long-standing concerns regarding its use in cancer patients.

Round 2
Reviewer 1 Report
Comments and Suggestions for Authors
The authors have corrected the various aspects that were pointed out. The result is a manuscript of better quality, and with greater academic solidity.
The bibliography has been significantly improved.
The current work can be published in the current format, unless another reviewer deems otherwise.
Author Response
Dear Reviewer, we sincerely appreciate your valuable contributions to the final version of our manuscript.
Reviewer 2 Report
Comments and Suggestions for Authors
In order for the final narrative of the manuscript to be honest to potential readers, two ideas should be included in the discussion:
1) Regarding the answer the authors have given us in question 2, in the last paragraph they say in the last sentence 'We chose not to perform Kaplan Meier test in this study group'. Previous explanations as to why they have not performed it can be incorporated into the discussion. We believe that this is the most prudent way to defend the results reported by the authors, especially if we think that some readers may criticise this point.
2) We think it is very good (as commented in answer 4) that the authors have added a sentence (lines 372-375) acknowledging the limitations of the sample size and the heterogeneity of the tumours considered. However, in order to demonstrate more honesty, it would be highly recommended to state explicitly that these findings should be considered as early preliminary results.
Comments on the Quality of English LanguageIt needs a revision from a native speaker.
Author Response
1) Regarding the answer the authors have given us in question 2, in the last paragraph they say in the last sentence 'We chose not to perform Kaplan Meier test in this study group'. Previous explanations as to why they have not performed it can be incorporated into the discussion. We believe that this is the most prudent way to defend the results reported by the authors, especially if we think that some readers may criticise this point.
Answer 1:
Thank you for your suggestion. We have incorporated the explanation into the Discussion section to clarify our decision not to perform the Kaplan-Meier test
2) We think it is very good (as commented in answer 4) that the authors have added a sentence (lines 372-375) acknowledging the limitations of the sample size and the heterogeneity of the tumours considered. However, in order to demonstrate more honesty, it would be highly recommended to state explicitly that these findings should be considered as early preliminary results.
Answer 2:
We have explicitly stated that our findings should be considered as early preliminary results to ensure transparency and clarity.
3) It was recommended that our manuscript undergo language editing
Answer 3:
Following your suggestion, we had our manuscript edited by MDPI English Editing. Based on their recommendations, we revised the title to ensure better fluency in English. All the changes made have been highlighted in yellow for clarity.